# Obesity and Depression: A Pathophysiotoxic Relationship

**DOI:** 10.3390/ijms262311590

**Published:** 2025-11-29

**Authors:** Francisco A. Monsalve, Barbra Fernández-Tapia, Oscar C. Arriagada, Daniel R. González, Fernando Delgado-López

**Affiliations:** 1Department of Preclinical Sciences, Faculty of Medicine, Universidad Católica del Maule, Talca 3466706, Chile; fmonsalve@ucm.cl (F.A.M.); oarriagada@ucm.cl (O.C.A.); 2Nursing School, Faculty of Health Sciences, Universidad de Talca, Talca 3465548, Chile; bfernandez@utalca.cl; 3Department of Basic Biomedical Sciences, Faculty of Health Sciences, Universidad de Talca, Talca 3465548, Chile; dagonzalez@utalca.cl; 4Biomedical Research Laboratory, Department of Preclinical Sciences, Faculty of Medicine, Universidad Católica del Maule, Talca 3466706, Chile

**Keywords:** obesity, depression, inflammation, HPA axis, microbiota-gut–brain axis, neuroinflammation, metabolic dysfunction

## Abstract

Obesity and depression are two of the most prevalent diseases with increasing trends worldwide; it has been some time since the first epidemiological associations were first described. Currently, there is abundant evidence showing the physiology and the molecular aspects that intersect the biology of both ailments. This narrative review aims to synthesize current evidence on the epidemiology and shared pathophysiology of obesity and major depressive disorder, emphasizing convergent inflammatory, neuroendocrine, metabolic, genetic, and gut–brain mechanisms. We aggregate evidence for a bidirectional relationship mediated by: (1) chronic low-grade inflammation (elevated CRP, IL-6, TNF-α; microglial activation); (2) HPA axis dysregulation (hyper/corticosteronemia, impaired feedback, altered CRH/ACTH signaling); (3) metabolic and neurotrophic signaling deficits (insulin and leptin resistance, dysregulated adipokines such as leptin/adiponectin, impaired BDNF and synaptic plasticity); (4) lipid-derived neurotoxicity and mitochondrial stress (saturated fatty acids, ceramides, oxidative stress); and (5) gut–brain axis perturbations (microbiota dysbiosis, increased intestinal permeability, LPS-driven endotoxemia, altered short-chain fatty acids and tryptophan–kynurenine metabolism). We highlight how these convergent pathways promote neuroinflammation and mood dysregulation in individuals with obesity and summarize clinical consequences for screening, integrated management, and targeted interventions that modulate immune, neuroendocrine, metabolic, and microbial processes. Finally, we outline priorities for identifying shared biomarkers and advancing personalized strategies via multi-omics and systems medicine to improve prevention, diagnosis, and treatment.

## 1. Introduction

Obesity and depression are two of the most prevalent diseases worldwide and represent significant problems, not only due to their direct impact on people’s health and well-being but also due to their socioeconomic repercussions [1]. Moreover, the global incidence of both pathologies continues to increase, and they are currently considered pandemics [2]. It is estimated that by 2030, more than 1 billion people will be obese, and depression will affect more than 350 million people worldwide [3]. These diseases occur in all age groups, but the likelihood of suffering from both pathologies increases in women, with pregnancy being the leading risk factor for developing both diseases [4,5,6,7]. Epidemiological evidence has identified strong associations between depression and obesity [1]. Although the etiopathogenic processes of both pathologies are perfectly distinguishable, growing evidence suggests a complex bidirectional relationship between obesity and depression, which could explain their parallel increase in the population. Major depressive disorder (296.2x or 296.3x according to DSM-5-TR, hereafter referred to as depression) [8] is associated with a higher risk of weight gain, and obesity, in turn, is associated with a greater vulnerability to developing depressive disorders [9]. Clinical and epidemiological data from studies conducted in the 1960s already showed an association between obesity and depression [10]. The incidence of depression in obese individuals is close to 30% [11], a higher rate than that observed in the general population. On the other hand, obesity is associated with a 1.18 to 5.25 times greater risk of developing depression [12], depending on the type of study and methodology used. Recent studies have found that obese people are 55% more likely to develop depression through their lifetime, while depressed individuals are 58% more likely to become obese than the general population [13].

Obesity and depressive disorders are closely related, but this relationship is likely multifactorial and complex, involving not only psychological and behavioral aspects but also shared biological mechanisms, which could explain the association between these two pathologies at different levels, including genetic, endocrine, immuno-inflammatory and metabolic mechanisms, along with the involvement of the vasculature and the intestinal microbiota.

This paper is a narrative review of recent scientific literature, focusing on the common pathophysiological mechanisms between obesity and depression. For this purpose, a search was conducted in databases such as PubMed, Scopus, and Web of Science, prioritizing articles in English and Spanish published until 2025. Relevant reviews and clinical and experimental studies addressing epidemiological, neuroendocrine, immunometabolic, and gut–brain axis aspects implicated in both pathologies were included. This review does not follow a predefined systematic protocol, but is aimed to integrate available knowledge from a comprehensive and up-to-date perspective.

## 2. Methodology

This narrative review was conducted to synthesize current evidence on the shared pathophysiological mechanisms linking obesity and depression. The search strategy was developed following the conceptual framework proposed by Preiss et al. (2013) [14], emphasizing inflammatory, neuroendocrine, and metabolic pathways.

A comprehensive literature search was performed in electronic databases, including PubMed, Scopus, Web of Science (WOS), SciELO, Cochrane Library, and Trip Database. The search covered publications in English and Spanish up to 2025, without geographical restrictions. The following keywords and Boolean combinations were used: obesity, depression, overweight, systemic inflammation, HPA axis, microbiota, gut–brain axis, neuroinflammation, and metabolic dysfunction.

Both experimental and clinical studies, as well as narrative and systematic reviews, were included when addressing epidemiological, neuroendocrine, immunometabolic, or gut–brain interactions relevant to the obesity–depression relationship. Articles were selected based on their methodological rigor, clarity of results, and contribution to the integrative understanding of the topic.

No predefined systematic protocol (e.g., PRISMA) was applied, as this review aimed to integrate and discuss emerging evidence from a broad and multidisciplinary perspective rather than to perform a quantitative meta-analysis. All information was critically analyzed and organized thematically to highlight biological mechanisms, clinical implications, and future research directions.

## 3. Epidemiology

### 3.1. Obesity (HP:0001513)

Obesity is a complex, multifactorial condition, characterized by overweight due to excess adiposity, which impairs health and quality of life. Its prevalence has risen markedly worldwide, driven by intersecting biological, behavioral, environmental, cultural and socio-economic determinants. The condition is associated with elevated risks of cardiometabolic diseases, certain cancers, and premature mortality [15], which are closely linked to obesity (HP:0001513) [16]. The World Obesity Atlas 2024 shows that the rising body mass index (BMI) of the general population is a global problem, being more pronounced in the poorest countries. This will cause economic losses exceeding $4 trillion by 2035, equivalent to approximately 3% of the world’s gross domestic product [17].

BMI (weight in kg/height in m^2^) is the most widely used formula to define overweight (BMI 25–29.9 kg/m^2^), obesity 1 (BMI 30–34.9 kg/m^2^), obesity 2 (BMI 35–39.9 kg/m^2^), or obesity 3 (BMI > 40 kg/m^2^) [18]. It is an easy-to-use anthropometric measurement formula used in health examinations and epidemiological surveys. The relationship between BMI and clinical outcomes has been extensively analyzed, and there is universal acceptance of BMI < 25 kg/m^2^ consistent with good health [19]. Global estimates suggest that 3.3 billion adults will have an elevated BMI by 2035, up from 2.2 billion in 2020. This represents an increase from 42% of adults in 2020 to over 54% in 2035. For children aged 5–19 years, the elevated BMI figure increases from 22% (430 million) to over 39% (770 million) by 2035 [17].

Due to their elevated BMI, these 770 million children are at increased risk of experiencing the early signs of noncommunicable diseases. By 2035, an estimated 68 million children will suffer from high blood pressure, about 27 million will develop hyperglycemia, and about 76 million will have low HDL cholesterol levels, all due to their high BMI [20]. These serious conditions progress asymptomatically, increasing the risk of developing chronic diseases in adulthood. Despite sporadic efforts to address this issue, without significant and coordinated action, obesity rates will continue to rise, and more people will die prematurely from obesity or obesity-related diseases. Furthermore, obesity-associated NCDs previously seen only in adults are increasingly common in children [17]. Reducing the prevalence of obesity worldwide and improving its management would generate significant benefits for health services, reducing countries’ economic expenditures related to the treatments of pathologies coexisting with obesity and decreasing hospitalizations due to these conditions, which translates into a better quality of life for the population [21]. Therefore, reducing obesity would not only improve individual health but also strengthen the efficiency of the entire health system, increasing the likelihood of achieving global goals to address NCDs in the population and ensuring better health for future generations [22].

### 3.2. Major Depressive Disorder (OMIM:608516)

Depressive disorders include a broad range of mood disorders. Among them, major depressive disorder (OMIM:608516) [16] (296.2x or 296.3x according to DSM-5-TR), a debilitating condition, was incorporated into the DSM-III in 1980 [23], being defined as a set of symptoms sustained over time that include apathy, abulia, sadness, hopelessness, anhedonia, anergy, sleep and appetite disorders, low self-esteem, and even suicidal ideation [24].

Approximately 350 million people worldwide suffer from depression [25]. This is about 4% of the population, including 5% of adults (4% in men and 6% in women) and 5.7% of adults over sixty years old. Depression is approximately 50% more prevalent in women.

The etiology of depression has multiple causes, including factors related to alcohol and drug abuse, marital and socioeconomic status, and comorbidities such as diabetes, cancer, HIV/AIDS, irritable bowel syndrome, autoimmune diseases, and obesity [13]. There is abundant evidence showing the connection between depression and NCDs, which masks its diagnosis and impedes its treatment, making it one of the most underdiagnosed and undertreated pathologies [26]. Although currently there are effective treatments for mental disorders, more than 75% of people in low-income countries do not receive treatment due to a lack of access to mental health services, a lack of trained personnel, and the associated social stigma of suffering mental disorders [27].

## 4. Pathophysiology of Obesity

### 4.1. Genetics

Information from the field of monogenic obesity has significantly contributed to the general knowledge about the genetics and physiology of body weight regulation. However, non-syndromic monogenic obesity represents only 5% of the obese population [28]. The heritability of BMI has been estimated to be between 40% and 70% based on twin, family, and adoption studies [29]. Most of the identified single-gene mutations are associated with the leptin-melanocortin pathway and may be present in up to 6% of individuals with severe obesity since childhood [30]. About 95% of individuals with obesity develop common polygenic obesity, which is multifactorial, and assessing its heritability is one of the main challenges today. Genome-wide linkage and association studies (GWAS) technologies have been instrumental in discovering genes involved in the complex etiology of human obesity [31].

One of the few promising genes that is linked to childhood obesity was the ectonucleotide pyrophosphatase/phophodiesterase 1 (*ENPP1*) gene, located on chromosome 6q [32]. Despite some inconsistencies in replicating this research, a meta-analysis involving 24,324 individuals supported the potential role of *ENPP1* in the pathophysiology of obesity [33]. Nevertheless, high-throughput genotyping techniques have allowed researchers to use GWAS to identify new loci associated with polygenic obesity in humans [31]. This has led to the uncovering of previously unknown genetic variants associated with obesity. Since the discovery of the genetic variant fat mass and obesity-associated gene (*FTO*) reported in 2007 [34], hundreds of loci carrying variants have been identified, including single nucleotide polymorphisms (SNPs) associated with measurable parameters in obesity, such as the *ADCY3* gene, which plays an important role in the regulation of body weight in humans [35]. Despite these loci being associated with obesity, GWAS methodology only explains 6% of the variation in BMI [36]. Although the high variability in BMI is one of the challenges for researchers, GWAS have demonstrated enormous potential in the discovery of new loci involved in the pathophysiology of obesity. In this sense, many of the identified loci involve genes that affect the neuronal circuits regulating appetite and satiety (*BDNF*, *MC4R* and *NEGR*) [31,37,38], energy and lipid metabolism (*FTO*, *RPTOR* and *MAP2K5*), insulin secretion and activity (*TCF7L2*, *IRS1*), as well as adipogenesis [39]. Many of these obesity-associated genes are also associated with other metabolic diseases such as diabetes, hypertension, and coronary artery disease [40]. Despite advances in the identification of loci associated with obesity, a large part of the heritability of body mass index (BMI) remains unexplained. The interactions between genetic and environmental factors, as well as epigenetic mechanisms, are still not fully understood. Furthermore, the clinical applicability of these findings for obesity risk prediction or treatment remains limited, as the identified genetic variants explain only a fraction of the total estimated heritability [36,39].

Beyond leptin–melanocortin pathways, genetic variation in the endocannabinoid system (ECS) represents a shared axis across metabolic and affective phenotypes. The *FAAH1* missense variant *C385A* (*rs324420*), which reduces enzyme stability and activity, elevates anandamide tone, with associations spanning cardiometabolic and neuropsychiatric domains [41]. Complementarily, *X*-linked *FAAH2* hemizygous loss-of-function variants have been described in humans presenting neurological features and metabolic disturbances, including obesity and hepatic steatosis [42]. On the receptor side, CNR1 (CB1; e.g., *rs1049353*) shows significant association with major depression [43], whereas evidence regarding CNR2 (CB2) is more nuanced, with no global case–control differences in panic disorder but a male-specific protective haplotype, alongside increased risk conferred by CNR1 variants in the same cohort [44]. Taken together, *FAAH1*/*FAAH2*–CNR1/CNR2 genetics align with reviews positioning the ECS in obesity pathogenesis and as a therapeutic target, supporting ECS-informed precision strategies for comorbid obesity–depression [45].

### 4.2. Neuroendocrine Dysregulation

Neuroendocrine dysregulation is common in obese patients. Neuroendocrine and immune systems are in constant communication, and alterations in one cause changes in the other, contributing to the comorbidities observed in obesity [9]. In general, obesity is associated with dysfunction of the hypothalamic–pituitary–adrenal (HPA) axis with a reduced response to cortisol feedback with the immune system (chronic low-grade inflammation) and dysregulation of metabolic pathways (leptin, insulin, NPY, and ghrelin, among others) [46].

The HPA axis is one of the most important neuroendocrine axes that plays a role in the regulation of the stress response via the secretion of glucocorticoids (GCs), cortisol in humans and corticosterone in animals [47], and which is hyperactivated, particularly in obesity. Under normal conditions, activation of the HPA axis suppresses the proinflammatory immune response, but in circumstances of danger or chronic stress, when the threatening stimulus remains over time, the HPA axis promotes an inflammatory response [48], a phenomenon known as glucocorticoid resistance, which in turn causes even more activation of proinflammatory immune cells [49]. Exposure to high cortisol levels can induce obesity by increasing appetite with a preference for high-calorie foods, stimulating visceral adipogenesis, and suppressing brown adipose tissue thermogenesis with a corresponding reduction in energy expenditure. Therefore, obese individuals with elevated cortisol levels are more prone to the metabolic consequences of obesity [50].

The association between cortisol levels and obesity is complex, as not all obese individuals have elevated cortisol levels, and the development of obesity coincides with an increase in factors that enhance cortisol production, such as chronic stress, chronic low-grade inflammation, consumption of sugar- and fat-rich foods, and reduced sleep [51]. On the other hand, hypercortisolism promotes the accumulation of visceral adipose tissue through the activation of the glucocorticoid receptor (GR) signaling cascade. This activity increases the production of adipose-derived proinflammatory cytokines, further stimulating the HPA axis and resulting in an attenuation of immune system activity [52]. The chronic inflammation typical of obesity can alter GR functioning, preventing the negative feedback generated by cortisol on the HPA axis, and proinflammatory cytokines activate elements of the cellular transduction cascades that prevent GR translocation to the nucleus or interfere with the interaction of GRs with response elements that promote gene activation [53], resulting in insensitivity to GC and loss of anti-inflammatory feedback [52]. Visceral obesity, together with the loss of muscle mass associated with hypercortisolism, favors the appearance of clinical parameters of the metabolic syndrome. However, obesity comorbidities are often defined by atypical features, and these are associated with normal to low cortisol levels. Individual variations in glucocorticoid sensitivity, which is partly genetically determined, may cause greater vulnerability, for example, to atypical depression in obesity, although the precise contribution of HPA activity remains to be established [54]. Although an association between the hypothalamic–pituitary–adrenal (HPA) axis and obesity has been described, their causality is controversial. Interindividual variability in glucocorticoid sensitivity complicates the interpretation of the findings, and there is disagreement as to whether hypercortisolism acts as a cause or consequence of the development of obesity. Despite the growing interest in this axis, consistent functional biomarkers that allow establishing personalized risk profiles are still lacking [55].

### 4.3. Peripheral Metabolic Dysregulation

The increase in adipose tissue associated with obesity is linked to changes in the pattern of adipokine secretion, causing a “metainflammation” in metabolically important tissues such as adipose tissue (AT), liver, skeletal muscle, pancreatic islets, and the brain [56]. Proinflammatory cytokines interfere with the insulin signaling pathway, leading to impaired glucose absorption and uncontrolled lipolysis, resulting in ectopic lipid storage, increasing insulin resistance in a vicious cycle [57].

A common consequence of the chronic low-grade inflammation observed in obesity is metabolic syndrome (MS), which is considered a collection of risk factors that predispose individuals to developing NCDs [17]. The diagnosis of metabolic syndrome is made when 3 of the following 5 risk factors are present: (1) central obesity (increased waist circumference, defined according to country-specific criteria); (2) high blood pressure (systolic > 130 mmHg, diastolic > 85 mmHg); (3) loss of glycemic regulation (high fasting blood glucose, blood glucose > 100 mg/dL); (4) low serum high-density lipoprotein (HDL) levels (<40 mg/dL in men and <50 mg/dL in women); (5) high serum triglyceride levels (>150 mg/dL) [56]. The presence of metabolic syndrome predisposes individuals to develop diabetes and cardiovascular diseases, as well as non-alcoholic fatty liver disease, obstructive sleep apnea, and cancer, among others. In fact, hyperleptinemia, hypoadiponectinemia, and insulin resistance (IR) are also widely linked to metabolic syndrome [58].

IR was initially proposed as the core of MS; however, obesity is now recognized as the triggering agent for each component of the syndrome, as many of the underlying mechanisms correspond to the deleterious effects of certain cytokines secreted by adipose tissue, potentially affecting the functioning of different organs [59]. In this sense, the excess of adipose tissue (perivisceral fat) accumulated in the abdominal region, unlike subcutaneous tissue, is metabolically active and releases several adipokines, such as TNFα, MCP-1, IL-6, TGF-β, PAI-1, and leptin, among others, which have been postulated to be the link between obesity and the pathophysiology of MS [60].

The excessive adiposity secretes a large amount of MCP-1, a cytokine that stimulates the recruiting and infiltration of circulating monocytes into the adipose tissue of obese individuals. These monocytes differentiate into M1 macrophages, which produce high levels of TNF-α [61], perpetuating the vicious cycle that gives rise to MS. TNF-α stimulates lipolysis, increasing the release of free fatty acids (FFA) into the circulation. The catabolism of FFA by skeletal muscle increases intracellular levels of acylCoA and diacylglycerol. These molecules are powerful allosteric activators of protein kinase C (PKC), which in turn phosphorylates the insulin receptor substrate 1 (IRS-1), altering the tyrosine kinase function of this receptor. This phosphorylation prevents the activation of the insulin receptor, which disrupts the intracellular signaling cascade, preventing the translocation of the glucose transporter GLUT-4 to the plasma membrane, resulting in hyperglycemia and the development of IR [62].

### 4.4. Peripheral and Central Inflammation

Along with neuroendocrine and metabolic dysregulation, chronic low-grade inflammation and increased susceptibility to immune-mediated diseases appear to be a key component of obesity, which is now considered a condition affecting the immune system [9]. Immune cells play a critical role in regulating adipose tissue phenotypes in response to physiological and pathophysiological stimuli [63,64]. Evidence that obesity produces chronic inflammation emerged in the 1990s, with the study by Hotamisligil et al., who showed increased levels of TNFα in the adipose tissue of obese rats [65]. Neutralizing TNF-α signaling improved insulin sensitivity, establishing a relationship between the immune response and metabolism. Currently, a large body of research indicates that chronic low-grade inflammation is the hallmark of adipose tissue dysfunction and systemic metabolic dysregulation [66].

Inflammation in adipose tissue plays a fundamental role in the development of obesity, depression, metabolic diseases, and other comorbidities. Furthermore, chronic low-grade inflammation represents the intersection of several molecular pathways that affect the immune system and the function of various organs through the secretion of proinflammatory cytokines [59], interfering with the expression of cell cycle regulatory proteins, apoptosis, or oxidative stress, leading to a high risk of developing other pathologies secondary to obesity [30].

In addition to white adipose tissue, skeletal muscle and the liver are also infiltrated by macrophages and T cells, which progressively accumulate in these tissues, enhancing their ability to secrete inflammatory mediators [67]. Obesity induces a phenotype shift in infiltrating macrophages in adipose tissue, from an anti-inflammatory M2 to a proinflammatory M1 state [68]. M1 macrophages are an important source of proinflammatory cytokines and, in adipose tissue, can be detected around dead or dying adipocytes, forming characteristic crown-like structures. The shift from the M1 to the M2 phenotype decreases adipose tissue inflammation and increases insulin sensitivity, in addition to reducing the metabolic complications observed in obesity [66], a shift promoted by lifestyle changes, particularly exercise. High levels of proinflammatory cytokines, in the absence of infection or tissue damage, are considered abnormal and tend to contribute to the development of cardiovascular diseases, diabetes, metabolic dysregulations, and neuropsychiatric diseases [13]. This peripheral inflammation, through humoral or neural pathways, especially via the vagus nerve, can induce an inflammatory brain state (neuroinflammation or central inflammation) (Figure 1). This type of inflammation creates a favorable environment for the development not only of metabolic diseases but also of neuropsychiatric diseases (degenerative or emotional and behavioral disorders) [9].

The central nervous system (CNS) maintains bidirectional communication with the immune system across the blood–brain barrier, which regulates the passage of nutrients, metabolites, and immune cells [69]. Under inflammatory conditions, the integrity of the barrier is disrupted, allowing the infiltration of plasma proteins, such as fibrinogen and immunoglobulins, as well as immune cells such as B lymphocytes, neutrophils, monocytes, dendritic cells, and T lymphocytes into the CNS parenchyma, further exacerbating neuronal damage. Metabolic diseases such as obesity, neurodegenerative diseases such as Parkinson’s or Alzheimer’s, and psychiatric diseases such as major depression, contribute significantly to the progression of neuroinflammation [70]. Within the brain parenchyma, glial cells such as microglia and astrocytes play a significant role in inflammation. Microglia cells function as resident macrophages, which react to damaging signals by releasing proinflammatory cytokines, whereas astrocytes regulate neuronal homeostasis and participate in the modulation of synaptic activity [71]. Both cells interact closely and contribute to synaptic plasticity under normal conditions, but also to dysfunction under inflammatory conditions [72]. Recent evidence highlights the role of cytokines in synaptic transmission and plasticity. IL-1β, IL-2, IL-6, IL-8, IL-18, INFα, INF-γ and TNF-α affect synaptic transmission and cognitive processes (memory and learning) in the hippocampal area [73].

### 4.5. Microbiota-Gut–Brain Axis

The microbiota-gut–brain (MBB) axis is a complex communication system mediated by hormonal, immunological, and neural signals between the gut and the brain [74]. The MBB axis comprises the CNS, enteric innervation, which includes extrinsic fibers of the autonomic nervous system (ANS) and intrinsic neurons of the enteric nervous system (ENS), the HPA axis, and the gut microbiota. The extrinsic innervation of the gastrointestinal (GI) tract connects the gut to the brain through vagal and spinal nerve fibers, while the brain connects the GI tract through sympathetic and parasympathetic efferent fibers [75]. Through this pathway the gut microbiota might impact neurodevelopmental processes and brain functions. Dysregulation of MGB axis communication is associated with metabolic diseases [76] and psychiatric disorders [77]. Furthermore, these alterations are frequently linked to changes in microbiota composition or function, potentially contributing to the disruption of the communication between the gut and the brain [74]. In humans, the highest abundance of microorganisms is in the distal intestine, with approximately 3 × 10^13^ microorganisms from more than sixty genera [78]. Although bacteria are the most abundant and best-studied microorganisms in the gut, a wide range of archaea, yeasts, single-celled eukaryotes, helminth parasites, and viruses are also included, although the role these microorganisms play in the MGB axis is still unknown [79].

Current evidence indicates that microbiome regulation of the CNS occurs primarily through neuroimmune and neuroendocrine mechanisms, often involving the vagus nerve [80]. This communication is mediated by molecules derived from microorganisms, including short-chain fatty acids (SCFAs), secondary bile acids (2Bas), and tryptophan metabolites [81]. These molecules act on enteroendocrine cells (EECs), enterochromaffin cells (ECCs), and the immune system of the intestinal mucosa. Some of these molecules can cross the intestinal barrier and enter the bloodstream and may even cross the blood–brain barrier [82]. Both barriers can be altered by the gut microbiome, stress, and inflammation, such as the chronic low-grade inflammation present in obesity, making them more permeable to the transit of immune cells, proteins, and cytokines [58]. Furthermore, endogenous metabolites capable of activating the CNS are produced by the gut microbiota, such as γ-aminobutyric acid (GABA), 5-HT, norepinephrine (NE), and dopamine (DA), although their true relevance to the responses they may exert on their regulatory role in the MGB axis is unknown [81].

The gut microbiota is the main source of endotoxins (lipopolysaccharide, LPS) and metabolites such as SCFAs, GABA, 5-HT, NE, and DA, all of which can contribute to chronic low-grade inflammation and affect neuronal signaling and excitability observed in obesity [83] (Figure 2). Changes in microbiota composition, such as those found in obesity dysbiosis, are associated with inflammation, metabolic dysregulation, and impaired mental health [84]. The mechanisms by which microbiota regulates brain function are still unclear; however, the inflammatory consequences of dysbiosis and its contribution to the metabolic and neuropsychiatric comorbidities of obesity is well documented [85]. Preclinical evidence on the impact of microbiota on affective behavior has grown rapidly, but its extrapolation to humans remains limited. The high interindividual variability in microbial composition, influenced by factors such as diet, medications, and environment, difficult to establish universal patterns. Furthermore, although multiple animal studies show clear effects of dysbiosis on mood, human results have been inconsistent due to methodological differences, confounding effects, and variable case classification [86].

## 5. Pathophysiology of Depression

One third of patients with depressive disorders are resistant to conventional treatments [87]. The need for new pharmacological targets, along with the difficulty of diagnosis, has led to the development of new theories about the origin of depression, in attempt to elucidate the pathophysiological processes involved in the development of this disease. New hypotheses have found common ground, complementing each other, leading towards an integrative theory.

### 5.1. Monoaminergic Hypothesis

Depression is a well-known psychiatric disorder that involves dysregulation of the monoamine system, leading to an imbalance of neurotransmitters such as NE, 5-HT, and DA [88]. Monoamines are involved in the process of information transmission through the connection of presynaptic and postsynaptic neurons [89]. Each monoamine binds specific receptors and exerts distinct functions in the brain [90]. Noradrenergic neurons extend from the brainstem to almost all brain areas, where NE modulates functions of the prefrontal cortex, working memory processing, and regulates behavior, attention, emotions, cognition, and social interactions [91]. Serotonergic neurons, are present in all brain areas. They are the largest cohesive neurotransmitter system in the brain, playing a crucial role in the regulation of appetite, circadian rhythm, anxiety, memory, and learning [24]. Finally, DA is important monoamine that drives motivation and modulates pleasure, reward, and emotion, as well as memory and attention [92].

The monoaminergic hypothesis origins in the 1950s, after the first pharmacological treatments for depression with iproniazid and imipramine, initially proposed for the treatment of tuberculosis and psychosis, respectively. These drugs shoed effects on mood that led to systematic clinical trials, demonstrating effective results as antidepressants [93,94,95]. Subsequently, in studies focused on elucidating the molecular mechanism of these compounds, the enzyme monoamine oxidase and the reuptake of 5-HT and NE were recognized as therapeutic targets, in addition to decreasing the reserves of these neurotransmitters in presynaptic neurons [96]. In the mid-1960s, the monoaminergic hypothesis was formulated due to the low activity of monoamines in the brain of patients with depression [97]. This hypothesis is based on the efficacy of antidepressant drugs such as selective serotonin reuptake inhibitors (SSRIs), norepinephrine-dopamine reuptake inhibitors (NDRIs), tricyclic antidepressants (TCAs), and monoamine oxidase inhibitors (MAOIs) [98]. The mechanisms of action of antidepressants for this hypothesis are: (1) inhibition of 5-HT and/or NE reuptake; (2) antagonistic presynaptic inhibition of 5-HT and/or NE; (3) inhibition of monoamine oxidase (MAO). The findings of these mechanisms of action showed that chronic treatment with antidepressants causes elevation of monoamines with consequent improvement in mood [90].

In addition to NE, 5-HT, and DA, γ-aminobutyric acid (GABA) has also been reported to be affected in patients with depression. GABA plays a role in depression and anxiety through its interaction with inflammatory cytokines, NF-κB, and the p38 MAPK signaling pathway [99].

### 5.2. Inflammatory Hypothesis

The inflammatory hypothesis shares common ground with the monoaminergic hypothesis, as these immune mediators affect neurotransmission systems as well [100] and stimulate kynurenine pathways, thereby decreasing the bioavailability of tryptophan for serotonin synthesis [101].

Therefore, chronic activation of the immune system causes sickness behavior, which shares symptoms with depressive disorders such as hyperthermia, nausea, loss of appetite, anhedonia, apathy, anorexia, sleep disturbances, fatigue, and loss of interest in social and physical settings [102]. In many neuropsychiatric or neurodegenerative diseases, the integrity of the blood–brain barrier is altered, allowing the infiltration of plasma proteins such as fibrinogen and IgG, in addition to immune cells such as B cells, neutrophils, monocytes, dendritic cells, and T cells into the CNS parenchyma, generating deleterious effects on CNS functioning and, therefore, clinical signs of the disease [70]. Cytokines secreted by immune cells act on the HPA axis, stimulating the production of corticotropin-releasing factor (CRF) and glucocorticoids resistance, perpetuating the production of inflammatory stimuli [103] (Figure 3). Stress also causes alterations in the intercellular junctions of the intestinal epithelium, allowing the passage of various molecules into the bloodstream, a phenomenon known as leaky gut [81]. This condition contributes to the pathophysiology of depression, as gut bacteria or its components can trigger an immune response, further aggravating the existing pathology. Supporting this theory, it has been observed that patients with depression have elevated serum levels of specific IgA and IgM against gut bacteria [104], and intestinal elimination with antibiotics restored the levels of inflammatory markers, even in the brain, after a stressful stimulus. Flinkkilä and cols. postulate that previous infections may play a crucial role in the pathophysiological processes of psychiatric illnesses [105].

Several studies support the relationship between depression and inflammation, with multiple points of convergence between both hypotheses (monoaminergic and inflammatory). This allows to consider hypotheses regarding adjuvant treatments to control proinflammatory mediators or enhance the anti-inflammatory capacity itself, that could show clinical improvements that antidepressants alone fail to achieve [90]. Although the monoaminergic and inflammatory hypotheses have provided fundamental explanations for the underlying mechanisms of depressive disorder, they fail to fully account for treatment resistance, which is present in approximately one third of patients. The interaction between neurotransmitters, immune mediators, HPA axis dysfunction, and alterations in neuroplasticity continue to be the subject of research. Therefore, it is essential to integrate complementary approaches such as the neurotrophic, metabolic, immunological, and those related to brain-microbiota communication and cellular metabolism hypotheses to achieve a more multimodal understanding of depressive disorder. This perspective is supported by a recent review that proposes that multiple molecular pathways converge in the pathophysiology of major depression: monoaminergic neurotransmission, chronic stress, neurotrophins, mitochondrial dysfunction, inflammation, microbiota-brain axis, among others [106].

### 5.3. Neurotrophic Hypothesis and Neurogenesis

Like the monoaminergic and inflammatory hypotheses, the neurotrophin hypothesis has played a key role in the pathophysiology of depression. Neurotrophins are a group of proteins essential for neuronal growth, survival, and differentiation [107]. The main neurotrophins in mammals are brain-derived neurotrophin factor (BDNF), nerve growth factor (NGF), neurotrophin-3 (NT-3), and neurotrophin-4 (NT-4) [108]. BDNF is vital for CNS neurogenesis, synaptic plasticity, neuronal development, survival, and maintenance.

BDNF is a neurotrophin that has been involved in the pathophysiology of depression [109]. BDNF and its receptor tropomyosin kinase B (TrKB) are engaged in different intracellular signaling pathways such as mitogen-activated protein kinase/extracellular signal-regulated protein kinase (MAPK/ERK), phospholipase Cγ (PLCγ), and phospho-inositide 3-kinase (PI3K) [110]. These pathways have an impact at the CNS level, particularly in the process of memory and mood regulation [111]. The ERK pathway is implicated in the regulation of mood and behavior in the depression model mediating the effects of antidepressant agents [112], while the PI3K signaling pathway is an important component in long-term potentiation, a key process in learning and memory [113]. Changes in BDNF levels in the CNS disrupt this signaling pathway, which could lead to several psychological disorders, including depression [90].

The process of neurogenesis in adults involves the generation of new neurons and neuronal connections in the dentate gyrus of the hippocampus and the lateral subventricular area [114]. Regarding the pathophysiology of depression, it has been proposed that there is a loss or reduction in the neurogenic capacity normally present in adults [115]. However, animal studies of reduced neurogenic capacity in the brains have shown that depression-like behavior does not necessarily follow a reduction in neurogenesis [116]. Consistent with the structural and functional changes in the hippocampus associated with experimentally induced depression, neurogenesis is essential for the structural and functional restoration of the hippocampus, which can be accompanied by improvements in depressive symptoms [117]. Moreover, many antidepressant drugs, such as SSRIs, MAO inhibitors, TCAs, ECTs, and mood stabilizers, have been shown to facilitate neurogenesis and also improve depression therapy [24] by significantly reducing ceramide levels (molecules in the brain that block cell growth), thereby increasing neurogenesis [118].

In depression and other psychiatric disorders, studies show alterations in BDNF activity, postulating it as the central element of the neurotrophic hypothesis. BDNF expression is decreased in depressive disorders, and antidepressant treatments may restore its expression [119]. It has even been proposed to evaluate plasma BDNF levels as a biomarker of depression, and ketamine, one of the most promising molecules as an antidepressant, would be able to modulate BDNF levels [120]. Likewise, proinflammatory cytokines or external immune stimuli such as LPS affect plasma BDNF levels, as well as BDNF expression in the CNS, causing depressive symptoms in those with inflammatory states [121].

## 6. A Pathophysiological and Toxic Relationship

The data supporting the role of inflammation in depression is extensive. Patients with depressive disorders exhibit all the cardinal features of an inflammatory response, including increased expression of proinflammatory cytokines and their receptors, as well as elevated levels of acute-phase proteins, chemokines, and soluble adhesion molecules in peripheral blood and cerebrospinal fluid (CSF) [122]. These molecules are postulated to be one of the possible links between depression and overweight/obesity, as they can generate a systemic inflammatory response that impacts the brain through multiple pathways, causing neuroinflammation and, consequently, brain dysfunction and alterations in neurotransmission [123].

### 6.1. Chronic Systemic Inflammation as a Central Link

Obesity and depression share a state of chronic low-grade inflammation, sustained by an overproduction of proinflammatory cytokines such as IL-1β, IL-6, and TNF-α (Figure 4). These cytokines not only affect peripheral metabolism but also access the CNS through humoral and neuronal pathways, promoting neuroinflammation [122]. This condition alters critical neurophysiological functions such as neuronal plasticity, neurogenesis, and synaptic transmission, particularly at the level of the hippocampus, prefrontal cortex, and amygdala, areas involved in mood regulation [9].

Activation of the peripheral immune system plays a prominent role in the pathophysiology of psychiatric and metabolic disorders and could be the main precursor to depression in obesity. Depression has been associated with elevated circulating levels of proinflammatory cytokines, chemokines, and adhesion molecules, as well as prostaglandins and arachidonic acid derivatives [122]. Individuals with depressed mood can be predicted by elevated blood levels of the transcription factor NF-κB, sympathetic activation, and glucocorticoid insensitivity, and those with anxiety disorders can be observed with elevated circulating levels of inflammatory markers such as CRP, IL-1β, IL-6, and TNF-α [51]. Individuals with obesity also present a chronic low-grade inflammation, which contributes to the development of metabolic and vascular disorders. Obese individuals who exhibit elevated levels of CRP are more likely to develop depression and anxiety [124]. CRP is a protein secreted by the liver in response to an increase in circulating proinflammatory cytokines, primarily IL-6 and, to a lesser extent, IL-1β and TNF-α. Studies highlight that CRP levels are higher in depressed patients, making elevated levels of this protein a predictor of the onset of depression in obese patients [125]. Dietary saturated fatty acids positively correlate with elevated CRP concentrations, contributing to systemic inflammation by promoting TLR4 signaling in macrophages. Furthermore, diet, as a serum index of inflammation, is closely related to emotional distress [126].

Several complementary mechanisms contribute to the progressive development of chronic low-grade inflammation associated with obesity. One of the main protagonists is white adipose tissue, as reported by findings that report associations between cytokine levels and measures of central adiposity [9] (Figure 5).

Nonetheless, it is important to note that healthy non-dysfunctional inflammation is necessary to maintain a healthy adipose tissue, as reported by several groups and studies [127] and similarly, a controlled healthy functioning inflammation has been shown to be important in maintaining proper neurological function and mood stability [128] aspects that are important to consider when studying groups of individuals that may not fit the expected phenotypes, like signs of depression or neurological disease associated with proinflammatory markers, or when studying borderline levels of particular inflammatory molecules that could be involved in low grade dysfunctional inflammation during healthy homeostasis.

### 6.2. Neurotoxic Effects of Lipids and Adipokines

Visceral adipose tissue, in addition to being an energy reservoir, acts as a proinflammatory endocrine organ that secretes adipokines such as leptin, resistin, and visfatin. In obesity, an excess of these molecules, along with the release of free fatty acids, induces oxidative stress and mitochondrial dysfunction in the brain. These effects can directly damage neurons or alter gene expression associated with synaptic plasticity, exacerbating the risk of developing depressive symptoms [122].

The production of proinflammatory cytokines by white adipose tissue is a major source of obesity-related inflammation. Excessive consumption of fats and sugars favors the accumulation of visceral adipose tissue, which contains a higher concentration of saturated fatty acids and is especially vulnerable to infiltration by immune cells compared to subcutaneous adipose tissue [129]. In accordance with this, the incidence of depression is higher when abdominal adiposity or waist-to-hip ratio is assessed with BMI [130]. The importance of excess adipose tissue for emotional deficits in obesity is highlighted in studies using rats, showing that prolonged feeding with high-fat diets that do not generate significant adipose deposition does not stimulate anxiety-like behaviors. However, in normal-weight individuals, these diets can enhance HPA axis activity, suppressing behavioral reward responses [131]. Under metabolic stress, adipocytes produce inflammatory mediators and chemoattractant molecules, such as monocyte chemoattractant protein-1 (MCP1), which can activate resident immune cells and recruit bone marrow-derived immune cells. These resident macrophages and T lymphocytes perpetuate the inflammatory situation in adipose tissue [15], and dietary saturated fatty acids may be a direct source of metabolic stress that can promote inflammation by increasing TLR4 signaling in adipocytes and macrophages. This activation stimulates the NF-κB and MAPK pathways, promoting the release of proinflammatory cytokines (TNF-α, IL-6), insulin resistance, and chronic low-grade inflammation. [132]. Adipose-derived inflammatory molecules, such as TNF-α, alter insulin signaling, promoting insulin resistance and impairing vascular health. Since these immunometabolic consequences increase the risk of depression, the expansion of adipose tissue and its adipokines would play a central role in the transmission of peripheral inflammation to the CNS, activating JAK/STAT and NF-κB pathways that disrupt tight junctions and increase permeability. These mediators allow the entry of proinflammatory signals that activate microglia and astrocytes, promoting neuroinflammation and neuronal dysfunction [56,133].

### 6.3. Gut–Brain Axis Dysfunction

Gut dysbiosis, a common feature in obesity, is associated with increased intestinal permeability, which allows the passage of endotoxins and bacterial components into the bloodstream. These can promote systemic immune responses, favoring neuroinflammation through the gut–brain axis. Furthermore, altered gut microbiota affects the synthesis of neurotransmitters such as serotonin, dopamine, and GABA, which are essential in the modulation of mood and behavior [74].

The gut flora contributes to energy balance by influencing nutrient absorption and controlling fiber fermentation. Microbiota is the main source of endotoxin (LPS), which is well known to contribute to systemic inflammation. Its specific composition is individual and derives from initial colonization at birth and its progression and stabilization from childhood to adolescence, which depends mainly on environmental factors. Several bacterial species produce metabolites that can impact physiology and health, such as short-chain fatty acids, or affect neuronal signaling and excitability in mood networks, such as serotonin and GABA [83]. Individual differences, the immune system, diet, infections, and antibiotic use are the main effectors of microbiota, but psychological stress can also modulate the gut flora [134]. Chronic changes in microbiota composition (dysbiosis) are associated with inflammation, insulin resistance, and mental health deficits [84]. Interestingly, both the obese phenotype and anxiety-like behavior are transmissible by transplantation of gut microbiota from diet-induced obese (DIO) animal models to normal-weight, germ-free mice. Furthermore, the microbiota of DIO mice can weaken tight junctions and trigger inflammation in both the gut and brain and promote brain insulin resistance [135]. A diet rich in saturated fatty acids and obesity can induce phylum-level shifts in the microbiota, from Bacteroidetes to Firmicutes, positively correlating with energy intake and CRP levels in obese children [136]. While shifts in microbiota composition have been reported in individuals with depression and may correlate with disease severity, the mechanisms by which the gut microbiota regulates brain function remain unclear; however, the inflammatory consequences of dysbiosis could contribute to the psychiatric and neurological comorbidities of obesity.

### 6.4. HPA Axis and Cortisol: Pathophysiological Mediator

The constant activity of the HPA axis impacts a wide spectrum of inflammatory processes, metabolic diseases, and psychiatric diseases. Stressful life events can trigger episodes of depression and anxiety, and stress at an early age represents a risk of developing clinical depression or anxiety disorders in adulthood [137]. The association between cortisol levels and obesity is complex, as not all obese people have elevated cortisol levels, but obesity does coincide with an increase in factors that produce cortisol, such as chronic stress, chronic low-grade inflammation, consumption of foods high in fat and sugar, and lack of sleep [138]. Chronic stress increases the secretion of adipose tissue-derived proinflammatory cytokines, which have a stimulating effect on the HPA axis, contributing to the attenuation of the immune response. It is also associated with downregulation of GC receptor gene expression, resulting in insensitivity to the action of cortisol and a loss of anti-inflammatory feedback [52]. However, comorbidity between depression and obesity is often defined by atypical characteristics, mainly associated with normal or low cortisol levels, primarily due to interindividual variations in genetically determined sensitivity to GCs, which in turn can cause greater vulnerability to depression in obese people [51].

Permanent activation of the HPA axis contributes to the comorbidities of obesity and mood, stimulating the intake of palatable and caloric foods. Dysregulation of this axis contributes to weight gain in stressed individuals through the action of cortisol, which stimulates brain GC receptors that reward the ingestion of these palatable and caloric foods. This effect transiently reduces HPA activity and relieves negative affective states for a brief time [139]. These observations are related to a reciprocal relationship between obesity and depression, in addition to being consistent with the interconnectedness between brain structures and the secretion of neurotransmitters that control mood, emotions, and motivation for food, ultimately causing body weight gain in those suffering from depression [140].

### 6.5. Clinical Implications

Understanding obesity and depression as interrelated conditions through immunometabolic, neuroendocrine, and microbial mechanisms has transformed the way these pathologies are addressed in clinical practice. Given this evidence, a dual diagnostic approach is essential, systematically assessing mental health in patients with obesity and, reciprocally, metabolic status in individuals with depression [9].

Traditional treatments focused exclusively on body weight or affective symptoms are insufficient to address the shared pathophysiological complexity. In this context, combined interventions that include anti-inflammatory strategies, cognitive-behavioral psychotherapy, stress management, HPA axis modulation, sleep improvement, and diet could represent a more effective clinical approach [4,51]. For more information on antidepressant treatments in obese patients with major depression, see the article “Adipose Tissue, Non-Communicable Diseases, and Physical Exercise: An Imperfect Triangle” in Section 6. Management of the Overweight-Obesity Patient [15].

Additionally, the use of probiotics and prebiotics, aimed at restoring intestinal eubiosis, has shown promising results in reducing depressive symptoms and inflammatory parameters in people with obesity, reinforcing the therapeutic role of the microbiota-gut–brain axis [74]. Likewise, some pharmacological agents under development seek to combine neuromodulatory effects with anti-inflammatory or immunoregulatory properties, proposing new therapeutic alternatives for patients with resistant depression and concomitant obesity [141].

Finally, these clinical implications require a multidisciplinary approach in which physicians, psychiatrists, psychologists, nutritionists, and internal medicine specialists work in a coordinated manner to achieve a comprehensive impact on patient health, thereby reducing both morbidity and public health costs.

## 7. Research Projections

The multiple pathophysiological pathways shared by obesity and depression open up new opportunities for translational research. A key area is the search for shared biomarkers capable of predicting the risk or progression of both conditions. For example, elevated plasma levels of IL-6, CRP, leptin, and cortisol could be integrated as part of an immunometabolic risk profile [13].

Furthermore, studies based on omics technologies such as genomics, epigenomics, transcriptomics, proteomics, metabolomics, and microbiome will allow for a more precise characterization of metabolic and neuropsychiatric phenotypes, facilitating the development of personalized therapies [39]. In addition, combined with the use of artificial intelligence, complex patterns in clinical and biological data that elude conventional analysis could be identified, helping in the identification of patient subgroups and design of targeted interventions. Another relevant line of research is the longitudinal study of cohorts from early life, aiming to determine whether inflammation, intestinal dysbiosis, or HPA axis dysfunction precedes the development of depressive or metabolic symptoms, or whether these conditions emerge simultaneously in response to adverse environmental factors.

Likewise, targeting the modulation of the neuroendocrine axis HPA by low-dose mifepristone (GR antagonism) in hypercortisolemic phenotypes; GLP-1 receptor agonists (semaglutide/tirzepatide) for weight loss, HPA attenuation, and mood benefit; as well as exercise training (aerobic + resistance) to increase BDNF and improve HPA feedback. Evaluate MC4R agonism or positive allosteric modulators to enhance POMC signaling (e.g., setmelanotide in indicated genotypes; investigational MC4R PAMs for broader use); augment α-MSH tone by reducing AgRP antagonism (nutritional protein/leucine, sleep optimization); support POMC neuron function via GLP-1R/GIPR dual agonists and leptin sensitizers.

Finally, research should pay special attention to the social and environmental determinants of health. Factors such as food insecurity, chronic stress, sedentary lifestyle, and exposure to environmental pollutants, such as endocrine disruptors, could act as common triggers that aggravate or perpetuate the connection between obesity and depression [2,4].

These research projections would allow us to improve the biological understanding of both diseases and to develop effective and accessible interventions that reduce their burden in the most vulnerable populations.

## 8. Conclusions

Obesity and depression, traditionally approached as independent clinical entities, share a complex network of pathophysiological mechanisms that explain their high comorbidity. Current evidence reveals that both pathologies are deeply interconnected through chronic inflammatory processes, neuroendocrine dysfunction, alterations in the gut–brain axis, and imbalances in the gut microbiota. This pathophysiological relationship not only explains the clinical overlap between the two conditions but also suggests that one could directly or indirectly contribute to the development and perpetuation of the other (see Figure 6).

This knowledge compels us to rethink diagnostic and therapeutic strategies. The clinical approach must be integrative, multidimensional, and personalized, considering both the biological factors and the psychosocial determinants that contribute to the obesity-depression cycle. This approach will require that appropriate technologies and expertise be brought together, and political will must support, guide and enable a multidisciplinary approach. Likewise, individual factors must be studied more precisely, identifying those that explain why not all obese individuals develop depression and vice versa. Furthermore, it is urgent to promote translational research that identifies shared biomarkers, new therapeutic targets, and preventive strategies with population impact. It is also important to determine whether the association between obesity and depression allows us to understand the also apparent correlation between low-weight individuals and a higher incidence of depression, when compared to the prevalence among normal-weight individuals, and whether this relationship is also bidirectional. This aspect should, in principle, have a different, if not negative, answer. For example, a vast amount of evidence on the correlation between obesity and depression is based on studies that associate dysregulation of adipokines and growth factors with adipogenesis and adipocytes, which are evidently decreased in underweight individuals. It should be noted that this part of the argument does not consider individuals whose weight loss is more directly associated with psychological disorders such as anorexia nervosa.

Addressing this relationship from a systemic perspective will improve treatment efficacy, reduce the burden of these diseases, and move toward a more comprehensive and person-centered care model. Recognizing and addressing this interdependence is not only a clinical necessity but also a healthcare responsibility in the context of the high prevalence of both conditions worldwide. This is part of the ongoing pandemics of mental health and metabolic diseases we are currently facing worldwide. There is also growing evidence that complex childhood post-traumatic stress disorders (CPTSD), or even more subtle yet unclassified versions, may play a determining role in both metabolic and mental disorders, opening up a range of possibilities when studying sensitivities associated with a specific genomic, proteomic, or epigenetic profile.

Eventually, the understanding of the mechanisms involved in regulating the neuroendocrine aspects of energy homeostasis will uncover new therapeutic targets to modulate not just purely metabolic disorders but also the pathways that directly affect cognitive decisions, in other words, pathways that control mood fluctuations that, more often than not, drive decisions relevant to a healthy or unhealthy energetic behavior, making us eat more or less, regardless of need, or making us exercise or not.

## Figures and Tables

**Figure 1 ijms-26-11590-f001:**
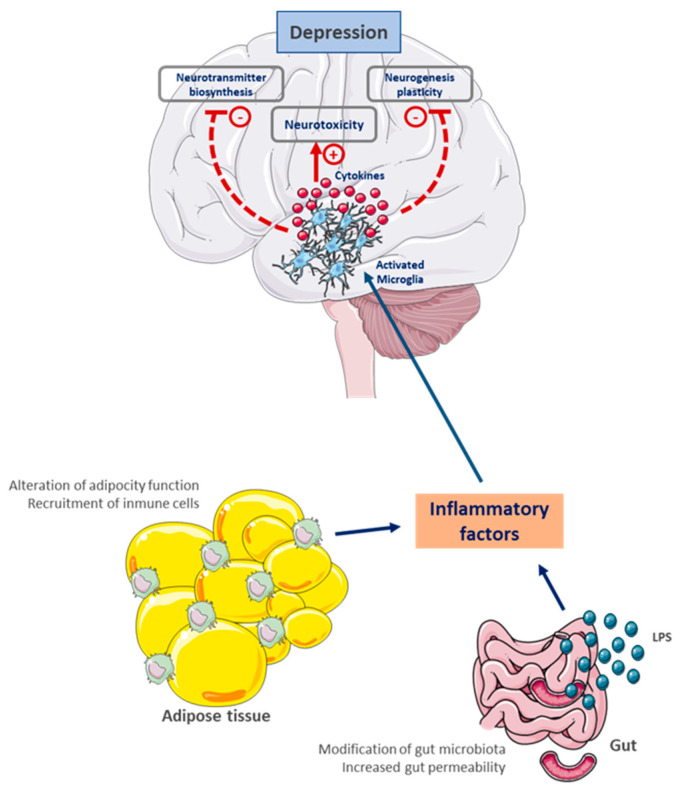
The relationship between systemic inflammation and depression through the gut-fat-brain axis. Altered adipose tissue and gut dysbiosis increase the release of inflammatory factors, such as cytokines and lipopolysaccharides, which reach the brain. There, they activate microglia, generating neuroinflammation and neurotoxicity. These responses decrease neurotransmitter synthesis, neurogenesis, and synaptic plasticity, contributing to the development of depressive symptoms. Thus, depression presents itself as a condition closely linked to peripheral and central inflammatory processes, where bidirectional communication between the body and the gut plays a crucial role in its pathophysiology. LPS, lipopolysaccharide; (+), activation; (−) inhibition.

**Figure 2 ijms-26-11590-f002:**
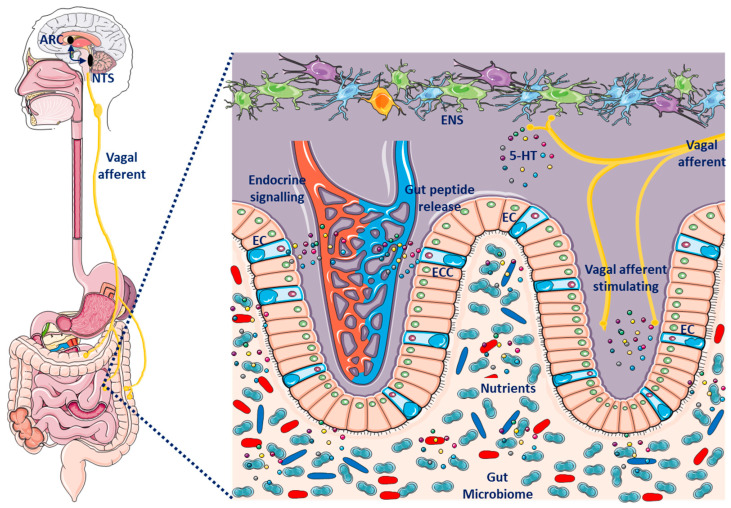
Major mediators of the gut–brain axis. Specialized intestinal epithelial cells, enteroendocrine cells (EECs), neuropod cells, and enterochromaffin cells (ECs), secrete gut peptides, including GLP-1, CCK, GIP, and PYY, on the basolateral side. These gut peptides are released in close proximity to vagal afferent neurons innervating the intestinal mucosa and activate these neurons. Vagal afferent neurons send signals to the nucleus tractus solitarius (NTS), which can send signals to higher-order brain regions, such as the arcuate nucleus (ARC). Vagal afferent neurons are also activated via the enteric nervous system, which can be activated by the release of gut-derived neurotransmitters, such as 5-HT, from ECs.

**Figure 3 ijms-26-11590-f003:**
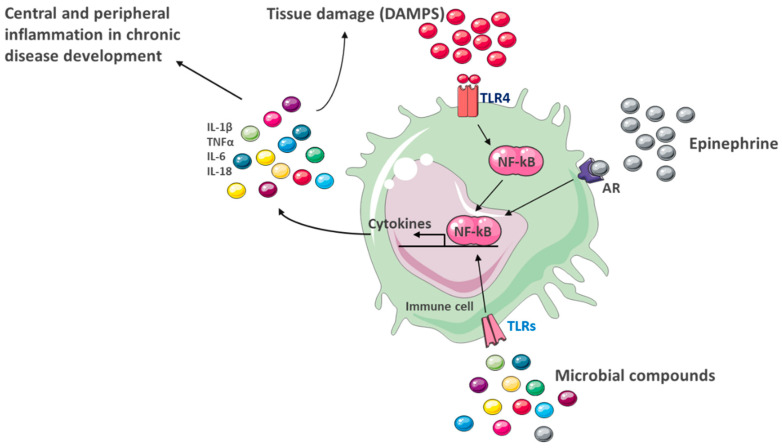
Multiple mechanisms may contribute to the dysregulation of cytokines in depression. The figure illustrates how various stimuli activate inflammatory responses through the transcription factor NF-κB in an immune cell. Microbial compounds and DAMPs bind to TLR receptors, while epinephrine acts on ARs, activating pathways that converge on NF-κB activation. This factor promotes the synthesis of proinflammatory cytokines such as IL-1β, IL-6, and TNF-α, which perpetuate inflammation. The release of these cytokines contributes to the development of chronic central and peripheral inflammation, linked to numerous metabolic, cardiovascular, and neuropsychiatric diseases. NF-kB pathways.; DAMP, danger-associated molecular pattern; AR, Adrenergic receptor; IL, interleukin; NF-kB, nuclear factor-kappa B; TLR, Toll-like receptor; TNF, tumor necrosis factor.

**Figure 4 ijms-26-11590-f004:**
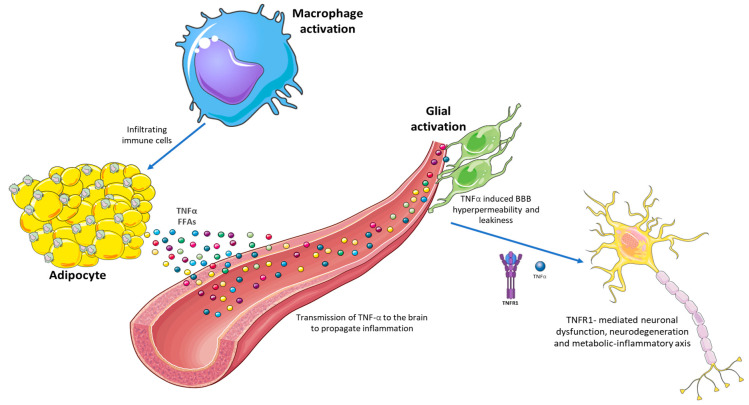
TNF-α activity in the central nervous system. The figure shows how, in obesity, adipocytes infiltrated by M1 macrophages stimulate the secretion of TNF-α, free fatty acids (FFAs) and other proinflammatory cytokines into the systemic circulation. These proinflammatory molecules, primarily TNF-α, disrupt the integrity of the blood–brain barrier, allowing these inflammatory mediators to enter the central nervous system. Exposure of neural tissue to these signals activates microglia and astrocytes, promoting a state of chronic neuroinflammation. This is associated with neuronal dysfunction and activation of the inflammatory-metabolic axis, processes that contribute to the development of depressive disorders linked to obesity.

**Figure 5 ijms-26-11590-f005:**
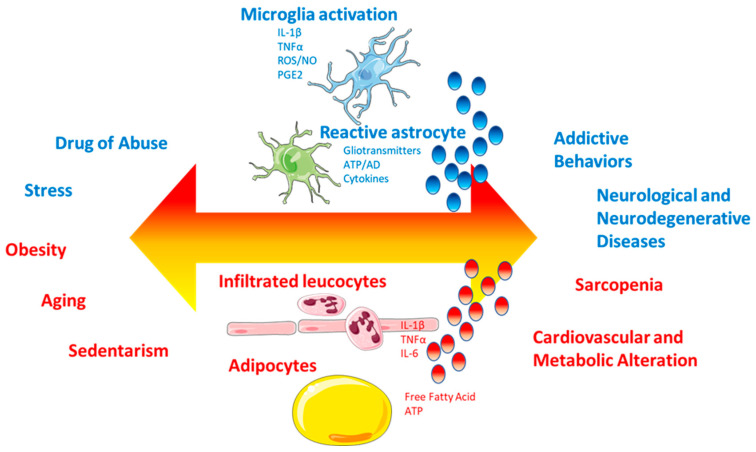
Role of central and peripheral inflammation in chronic disease development. Lifestyle factors such as drug consumption, stress, obesity, sedentary behavior, and conditions such as aging have been implicated in the development of chronic inflammation, a key driver of several chronic diseases. IL, interleukin; TNF, tumor necrosis factor; ATP, adenosine triphosphate; AD, adenosine; ROS, radical oxygen species; NO, nitrogen oxide; PGE2, prostaglandin E2.

**Figure 6 ijms-26-11590-f006:**
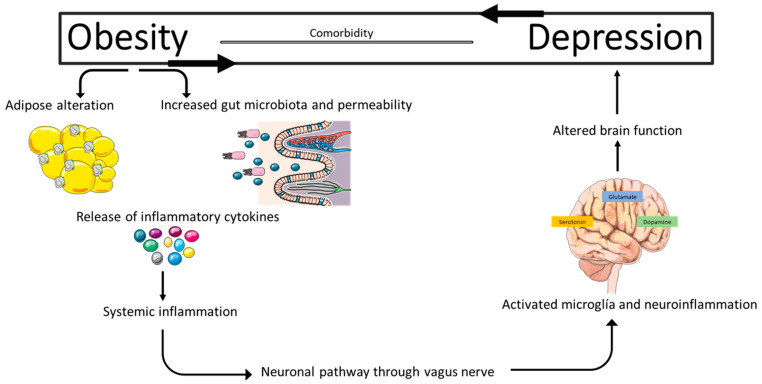
Pathophysiology of the comorbidity between obesity and depression. The figure illustrates the interaction between adipose tissue, the gut, and the brain in the context of chronic inflammation and depression. Adipocyte dysfunction and alterations in the gut microbiota increase the release of proinflammatory cytokines and lipopolysaccharides (LPS), which cross the intestinal barrier and promote activation of the immune system. These inflammatory mediators reach the brain, where they affect the neurotransmission of serotonin, dopamine, and glutamate, disrupting neurochemical balance. In turn, brain dysfunction can reinforce peripheral inflammatory processes, perpetuating a vicious cycle between metabolism, inflammation, and mood.

## Data Availability

No new data were created or analyzed in this study. Data sharing is not applicable to this article.

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
