# Peer review of "Obesity and Depression: A Pathophysiotoxic Relationship"

_ijms, 2025, doi:10.3390/ijms262311590_

Round 1

Reviewer 1 Report

Comments and Suggestions for Authors

This is an excellently written and well-organized review that addresses the complex and clinically significant relationship between obesity and depression. The manuscript demonstrates a strong background of the relevant literature, providing a balanced and comprehensive overview of the field. The structure and the arguments are coherent, and the progression of ideas is easy to follow. The English language is clear and precise, with a professional.

Overall, this is a well-rounded, insightful, and high-quality review that will be of great interest to researchers working in the field of metabolic and neuropsychiatric disorders.

Specific Comments

1. Future Perspectives:

The manuscript would benefit from a more explicit section highlighting future research directions. While the current synthesis is thorough, adding a concise paragraph outlining the potential therapeutic implications and emerging strategies that may simultaneously target both obesity and depression would strengthen the review’s translational relevance.

2. Therapeutic Approaches:

The authors could further elaborate on plausible therapeutic approaches—for example, interventions that integrate metabolic and neurobehavioral pathways, or pharmacological targets that address shared molecular mechanisms underlying both disorders. Including a conceptual model or schematic illustration could further enhance reader understanding.

3. Hypothesis Integration:

Some of the authors’ proposed hypotheses are insightful and could be more clearly incorporated into the narrative. Integrating these ideas in the concluding sections, framed as potential testable hypotheses, would make the review more engaging and forward-looking.

Reviewer 2 Report

Comments and Suggestions for Authors

This is an interesting and -in my opinion- quite clearly written article on the shared pathological mechanisms underlying obesity and depression. 

I would suggest some minor changes: 

-page 3, 2.2 Depression: "Depressive disorders include a broad range of mood disorders." It seems reasonable for the authors of the article to clearly define what they mean by the term “depression.” In addition, it would be advisable to refer to current diagnostic criteria (DSM-5-TR or ICD-10/ICD-11)

-In my opinion the figures could be described in more detail to make them more self-descriptive

- Antidepressants often do not have a beneficial effect on patients' body weight, so it would be reasonable to shortly discuss potential pharmacological methods for obese patients with depression (as clinical implications of such research)

Reviewer 3 Report

Comments and Suggestions for Authors

Summary

This narrative review explores the correlation between obesity and depression, focusing on shared pathophysiological mechanisms. The topic is relevant, and the manuscript has potential. However, improvements in scientific framing and methodological transparency is necessary. References must be adjusted to MDPI formatting standards. Additionally, the manuscript lacks numbers of lines, which is essential for peer review and should be added upon revision.

Abstract

The abstract is currently vague and non-specific. It should: clearly state the aim of the review; Mention the main shared biological pathways discussed.

Introduction

Human Phenotype Ontology (HPO) terms should be explicitly referenced when phenotypic traits are mentioned.

The correlation between obesity and depression should be critically synthesized investigating on comprehensive meta-analyses or large cohort studies. For example, the systematic review https://doi.org/10.1111/obr.12052 provides robust evidence linking both conditions with behavioral, psychological, and lifestyle factors.

The statement regarding age and sex differences (ref: Dębski et al., 2024) should be expanded and supported with at least one or two additional large-cohort studies.

Although this is a narrative review, a dedicated subsection titled "Search Strategy" should be included to describe how sources were selected. This will significantly strengthen methodological transparency.

Section 2.1

The opening sentence should be reformulated for clarity and better academic tone. The current phrasing is overly general.

Section 2.2

Depression should be defined according to DSM-5 diagnostic criteria (PMID: 24986345) to ensure clinical precision.

Section 3.1

For the GWAS data discussed, please include (if present) rsIDs of the associated SNPs as  well as additional details about the chromosomic position of those variants.

Sections 3–4

The genetic section would be significantly strengthened by including genes and variants linked to the endocannabinoid system, which represents a shared molecular axis in both obesity and depression. Suggested references: FAAH2 (https://doi.org/10.1016/j.gene.2025.149703); FAAH1 (https://doi.org/10.1002/biof.1911); CNR1 (https://doi.org/10.1016/j.phrs.2010.01.002); CNR2 (https://doi.org/10.1080/10615806.2020.1732358).

Including a genes-functions summary table might provide strong added value.

Section 5

This section is clear. Nevertheless the TNF-α pathway should be expanded with a diagram/figure showing cytokine overproduction mechanisms in obesity and its interaction with neuroinflammatory signaling.

Round 2

Reviewer 3 Report

Comments and Suggestions for Authors

Authors addressed all the reviewer's comments. References must be formatted according to the MPDI's Instruction for Authors.